# Selection and Validation of Reference Genes for qRT-PCR Analysis in the Oil-Rich Tuber Crop Tiger Nut (*Cyperus esculentus*) Based on Transcriptome Data

**DOI:** 10.3390/ijms22052569

**Published:** 2021-03-04

**Authors:** Xue Bai, Tao Chen, Yuan Wu, Mingyong Tang, Zeng-Fu Xu

**Affiliations:** 1CAS Key Laboratory of Tropical Plant Resources and Sustainable Use, Xishuangbanna Tropical Botanical Garden, Innovation Academy for Seed Design, Chinese Academy of Sciences, Menglun, Mengla 666303, China; baixue2015@xtbg.ac.cn (X.B.); chentao@xtbg.ac.cn (T.C.); yangyunju@xtbg.ac.cn (Y.W.); 2College of Life Sciences, University of Chinese Academy of Sciences, Beijing 100049, China; 3Center of Economic Botany, Core Botanical Gardens, Chinese Academy of Sciences, Menglun, Mengla 666303, China; 4State Key Laboratory for Conservation and Utilization of Subtropical Agro-Bioresources, College of Forestry, Guangxi University, Nanning 530004, China

**Keywords:** *Cyperus esculentus*, tuber, reference gene, qRT-PCR, gene expression, normalization, RankAggreg

## Abstract

Tiger nut (*Cyperus esculentus*), a perennial C_4_ plant of the *Cyperaceae* family, is an unconventional crop that is distinguished by its oil-rich tubers, which also possesses the advantages of strong resistance, wide adaptability, short life periods, and large biomass. To facilitate studies on gene expression in this species, we identified and validated a series of reference genes (RGs) based on transcriptome data, which can be employed as internal controls for qRT-PCR analysis in tiger nut. Fourteen putative candidate RGs were identified and evaluated across nine different tissues of two cultivars, and the RGs were analyzed using three different algorithms (geNorm, NormFinder, and BestKeeper). The stability rankings of the candidate RGs were merged into consensus lists with RankAggreg. For the below-ground storage organ of tiger nut, the optimal RGs were *TUB4* and *UCE2* in different developmental stages of tubers. *UCE2* and *UBL5* were the most stably expressed RGs among all tissues, while *Rubisco* and *PGK* exhibited the lowest expression stability. *UCE2*, *UBL5* and *Rubisco* were compared to normalize the expression levels of the *caleosin* (*CLO*) and *diacylglycerol acyltransferase 2-2* (*DGAT2-2*) genes across the same tissues. Our results showed that the RGs identified in this study, which exhibit more uniform expression patterns, may be utilized for the normalization of qRT-PCR results, promoting further research on gene expression in various tissues of tiger nut.

## 1. Introduction

To pursue a better quality of life and a healthier diet for humans, many economically important crops have been intensively studied. Meanwhile, an increasing number of agricultural sources of cooking oil have emerged to meet the increasing market demand. Generally, the majority of plants accumulate abundant oil in their seeds (e.g., soybean, rapeseed, peanut, and sunflower), as well as their mesocarps (e.g., olive, oil palm, and avocado), while a smaller number of oil-rich plants accumulate oil in non-seed vegetable tissues [1,2,3,4,5]. A wild tomato species (*Lycopersicon pennellii*) accumulates lipids on the epicuticle of leaves [6]. The oil-storage organs of *Tetraena mongolica* are phloem cells and xylem parenchyma of stems [7]. Tiger nut (*Cyperus esculentus*) is a perennial herb of Cyperaceae that is widely distributed in almost all middle- and low-latitude areas [8]; this plant stockpiles oil in its tubers as an underground storage organ [9]. Tiger nut is an unconventional crop that is distinguished by its oil-rich tubers, and it is also known as yellow nutsedge and chufa. Oleic acid (C18:1) is abundantly contained in the tuber oil of this plant and is a kind of monounsaturated fatty acids (MUFAs) [10], which has positive effects on human health, especially by reducing the blood lipids [11], blood pressure [12], and the incidence of type 2 diabetes mellitus [13]. At present, the majority of tiger nut researchers have primarily focused on the cultivation technology [14], oil extraction process [15,16], analysis of nutritional components [17] and medicinal values [18], and development of tiger nut milk [19,20]. However, few studies have undertaken gene expression analyses and functional analyses of this species [21,22,23]. To facilitate the studies of new genes and pathways related to lipid accumulation in tiger nut, it is necessary to select optimal reference genes (RGs).

Gene expression analysis is a widely employed and powerful method for elucidating the complex regulatory networks of the genetic, signaling, and metabolic pathway mechanisms that are active during the plant life cycle [24]. Quantitative real-time reverse transcription-polymerase chain reaction (qRT-PCR) has become the most popular method for quantitative gene transcription analysis, and this technique has many advantages, such as high sensitivity, accuracy, and specificity [25]. A prerequisite for the reliable analysis of gene expression is the normalization of qRT-PCR data, which can be utilized to minimize the nonspecific variations caused by variations in the quantity and quality of mRNA and variations in the efficiencies of reverse transcription and PCR [26]. As internal controls, appropriate RGs are necessary to normalize gene expression and, thus, avoid errors caused by various experimental procedures analyzing different plant tissues and plants in different developmental stages.

In general, most of the previously reported RGs are housekeeping genes, such as *actin* (*ACT*) [27,28], *tubulin* (*TUB*) [29], *polyubiquitin* (*UBQ*) [30], *elongation factor 1-α* (*EF1-α*) [31], *glyceraldehyde-3-phosphate dehydrogenase* (*GAPDH*) [32,33], and *ribosomal RNAs* (*18S rRNA* or *28S rRNA*) [34,35]. If only a common internal RG is used based on literature reports, it may lead to not only inaccurate test results but also incorrect conclusions [26]. Therefore, researchers first must carefully select RGs from a large number of RGs to suit their respective experimental conditions.

To identify suitable RGs for accurate quantification of target genes in tiger nut, we selected 14 RG candidates according to transcriptome data, namely, *18S ribosomal RNA* (*18S*), *actin* (*ACT*), *actin-depolymerizing factor 7* (*ADF7*), *elongation factor 1-alpha* (*EF1α*), *elongation factor 2* (*EF2*), *glyceraldehyde-3-phosphate dehydrogenase* (*GAPDH*), *ribosomal protein L11* (*RPL11*), *ribulose bisphosphate carboxylase/oxygenase* (*Rubisco*), *ubiquitin-conjugating enzyme 2* (*UCE2*), *ubiquitin-like protein 5* (*UBL5*), *cyclophilin* (*CYC*), *phosphoglycerate kinase* (*PGK*), *malate dehydrogenase* (*MDH*), and *tubulin beta-4* (*TUB4*). The expression stability of the selected candidates was evaluated by three different algorithms using a set of cDNAs obtained from different samples of two tiger nut cultivars. Additionally, to verify the expression stability of candidate RGs, the relative expression levels of *caleosin* (*CLO*) and *diacylglycerol acyltransferase 2-2* (*DGAT2-2*) in different tissues were normalized by the most stable and least stable RGs and compared with the expression levels revealed by the transcriptome sequencing.

## 2. Results

### 2.1. Primer Specificity and PCR Amplification Efficiency

Fourteen candidate RGs (*18S*, *ACT*, *ADF7*, *CYC*, *EF1α*, *EF2*, *GAPDH*, *MDH*, *PGK*, *RPL11*, *Rubisco*, *TUB4*, *UCE2*, and *UBL5*) and two target genes (*CLO* and *DGAT2-2*) of tiger nut were selected for qRT-PCR normalization. Further details for these genes are presented in Table 1. Melting curve analysis is a means of assessing the dissociation characteristics of double-stranded DNA during heating [36]. The specificity of each primer set was validated by melting curve analysis. For all primer sets, the melting curve exhibited a single amplification peak (Figure 1). The qRT-PCR efficiency for all 14 candidate RGs ranged from 92.54% to 118.22%, and the correlation coefficients varied from 0.9914 to 0.9998 (Table 1).

For each primer pair, a single amplicon of the expected size was obtained via RT-PCR analysis (Appendix A) of the cDNA template prepared from total RNA. Next, the amplified products were column-purified and cloned into the pGEM^®^-T Easy cloning vector (Promega Corporation, Madison, WI, USA) in preparation for sequencing. Sequencing results showed that the targeted region of each target transcript had been successfully amplified by RT-PCR, with all sequences returning 100% homology to the region targeted for amplification for each assessed candidate RG.

### 2.2. Threshold Cycle (C_t_) Values of Candidate RGs

To assess the expression stability of candidate RGs across all experimental samples, the transcript abundances of the 14 candidate RGs were evaluated based on mean C_t_ values. The average C_t_ values for these candidate RGs ranged from 8 to 26, with most values falling between 18 and 22 across all samples. *CYC* had the lowest expression level with a C_t_ value as high as 25.84 cycles, while *18S* was the most abundantly expressed gene with a C_t_ value as low as 8.40 cycles (Figure 2). The C_t_ values of *MDH* (24.64 ± 0.57) in samples from Xinjiang (XJ) and *UCE2* (22.48 ± 0.44) in samples from Yunnan (YN) with the minimum SD indicated that these genes were the most stable genes in the two cultivars. Moreover, *ADF7* (21.46 ± 0.48) in samples of YN and *UCE2* (24.10 ± 0.65) (Figure 2) in samples of XJ were the next most stable genes. Additionally, the most unstable gene was *Rubisco* (17.53 ± 2.60) in the XJ cultivar and *PGK* in the YN cultivar (21.50 ± 2.41) (Figure 2 and Appendix A). The results also showed that *ADF7* and *UCE2* exhibited the smallest variation in expression levels, whereas *Rubisco* and *PGK* displayed the largest variation among the tissues of the two cultivars.

To analyze the stability of 14 RGs more accurately, several software programs, including geNorm [26], NormFinder [37], and BestKeeper [38], were utilized for statistical analysis. Each program evaluated the stabilities of these candidate RGs and ranked them from the most stable to the least stable. The data of each RG from different samples were analyzed separately and subsequently integrated by RankAggreg (a package of R software) [39]. To perform a comprehensively analysis, two different cultivars of tiger nut, named “YN” and “XJ”, were collected, and the 14 candidate RGs were evaluated in four experimental sets consisting of samples collected at defined developmental stages. The first experimental set included young leaves, mature leaves, leaf sheathes and stem apexes, collectively known as the aboveground group. The second experimental set consisted of roots, rhizomes and tubers in three developmental stages, collectively known as the underground group. Moreover, tubers in three development stages comprised the third set. The last set consisted of all of the abovementioned plant tissues, known as total. The tissues of the YN and XJ tiger nut cultivars are shown in Appendix A, respectively. Afterwards, the expression stability of candidate RGs of the two cultivars of tiger nut was analyzed and compared using different software programs.

### 2.3. Expression Stability of Candidate RGs

#### 2.3.1. GeNorm analysis

GeNorm was mainly used to select the most stable RG by calculating the gene expression stability measure (*M*) based on the average pairwise expression ratio [26]. The lower the *M*-values of the candidate gene are, the more stable it is [40]. The *M*-values of 14 candidate RGs calculated by geNorm were all less than 1.5 except *Rubisco* in the whole tissues (Table 2). Therefore, all of these candidates meet the basic requirements of RGs, except *Rubisco*. For the YN tiger nut cultivar, *UCE2* (*M* < 0.403) was the most stable RG in all groups; the second most stable RGs were *MDH* (*M* < 0.409 in aboveground and total tissues) and *TUB4* (*M* < 0.062 in underground and tuber tissues) (Table 2). A similarity between two cultivars of tiger nut was that *UCE2* (*M* < 0.267) was the most stable RG in aboveground, underground, and total tissues. There were a number of differences between the two cultivars. The optimal RGs in the tuber development stages of XJ were *TUB4* (*M* = 0.220) and *UBL5* (*M* = 0.241), and another difference was that *UBL5* (*M* < 0.292) and *ADF7* (*M* < 0.268) were the second most suitable RGs in aboveground and total tissues, respectively (Table 2). Generally, *UCE2* was the best choice in the geNorm rank list. Conversely, the most unstable gene was *Rubisco*.

As a single RG may not be sufficient to accurately quantify the level of gene expression, it is necessary to select two or more RGs to ensure reliable normalization [26]. GeNorm provides pairwise comparison (Vn/Vn + 1) analysis to determine the optimal number of RGs. The value of Vn/Vn + 1 was less than 0.15, and the number of optimal internal RGs was *n*. The pairwise comparison analysis of two tiger nut cultivars indicated that the optimal number of RGs for normalization of qRT-PCR data was two (Figure 3). Therefore, a combination of *UCE2* and *MDH* was optimal for the YN tiger nut cultivar, and the combination of *UCE2* and *ADF7* was optimal for the XJ tiger nut cultivar among the total samples, as determined from the results obtained by geNorm (Table 2).

#### 2.3.2. NormFinder Analysis

To further confirm the stability of the RGs obtained by geNorm, we applied NormFinder software to identify the optimal normalization genes among the candidates [37]. Using the NormFinder algorithm, the stability value (*M*) of each candidate RG in each tissue group was calculated and listed (Table 3). This algorithm was similar to geNorm in that it could calculate different stable values and sort them. The difference was that NormFinder selected only one optimal gene, while geNorm selected two or more genes. In particular, *UCE2* and *UBL5* appeared most frequently in the normFinder rank of the two tiger nut cultivars.

The stability list obtained by normFinder was slightly different from that obtained by geNorm. For the YN cultivar, *UCE2* (*M* < 0.359) and *UBL5* (*M* < 0.409) were the most stable in aboveground and total tissues, while *TUB4* (*M* < 0.221) and *RPL11* (*M* < 0.290) were the top two RGs in both underground tissues and tubers. For the tissues of XJ, the top RGs were the same as with YN, while the second most stable genes were different. *UCE2* (*M* = 0.233) and *EF1α* (*M* = 0.292) were optimal in aboveground tissues; *TUB4* (*M* = 0.190) and *UBL5* (*M* = 0.233) were the best in underground tissues; the most stable RGs were *TUB4* (*M* = 0.191) and *PGK* (*M* = 0.241) in tubers; and *UBL5* (*M* = 0.231) and *UCE2* (*M* = 0.345) were the top two RGs of all samples. In summary, *UCE2*, *UBL5*, and *TUB4* appeared most frequently in the optimal RGs. *Rubisco*, *PGK* and *ADF7* were the most unstable candidates in all sets, as determined by NormFinder.

#### 2.3.3. BestKeeper Analysis

BestKeeper is an Excel-based tool that evaluates the stability ranking of RGs based on the coefficient of variance (CV) and the standard deviation (SD) of the average C_t_ values [38]. The most stable gene exhibited the lowest of these two values, and genes with SD higher than 1.00 were considered to be unacceptable and were excluded. The removed genes included *EF1α*, *TUB4*, *Rubisco*, and *PGK* in the YN group and *PGK*, *ACT*, *RPL11*, *CYC*, *TUB4*, *GAPDH*, *EF2*, and *Rubisco* in the XJ group (Table 4).

For aboveground tissues, *MDH* and *CYC*, as well as *MDH* and *EF2*, with the lowest SD and CV values, were the top two ranks corresponding to YN and XJ tiger nut cultivars, respectively. For the underground group, the top two candidates of YN and XJ tissues were *UCE2* and *MDH*. By comparison, it was observed that there were some differences in the expression levels of candidate genes between aboveground and underground tissues. Moreover, the optimal RGs were consistent in the tuber tissues of the two cultivars: *UCE2* and *TUB4*. For total tissues, *UCE2* and *MDH*, as well as *UCE2* and *18S*, were the most stable candidates in YN and XJ tiger nut cultivars, respectively, which exhibited the smallest SD and CV values. Throughout the ranks of BestKeeper, *UCE2* and *MDH* appeared most frequently in the top two, exhibiting the best stability among all groups.

### 2.4. Unified Rank Lists by RankAggreg

Three different algorithms were used to analyze the expression stability of 14 potential RGs in nine experimental tissues of two cultivators. The results of the aforementioned methods produced too many ranks to analyze. It was necessary that a new algorithm be introduced to unify these results. RankAggreg is an R package for the ranking aggregation algorithm [39], and this software was the best choice to merge these ranking lists in different categories.

RankAggreg integrated rank lists of 14 reference genes in the two cultivars YN and XJ, as well as among four different tissue groups, including aboveground, underground, tuber and total samples (Table 5). The most stable candidate genes were consistently *TUB4* in the tubers and *UCE2* in the other three groups of two cultivars. In particular, *TUB4* was the most stable gene in the tubers of the two cultivars but was low-ranked in the other groups, indicating that *TUB4* was a tuber-specific stable gene. In addition, the most unstable genes were *Rubisco* and *PGK* in the majority of tissues.

However, the intermediate orderings of each group were distinct. For example, the rankings of *ADF7* varied considerably, as this gene was ranked second to last in YN underground tissues but fourth in XJ underground tissues. Similarly, *PGK* also displayed a higher diversification, as it ranked No. 2 in XJ tubers but No. 7 and Nos. 10–13 in other groups (Table 5). The analysis of two cultivars can remove differences between cultivars to some degree. For a clearer comparison of the stability rankings obtained by different software, all ranking orders of 14 reference genes were integrated by geNorm, NormFinder, BestKeeper and RankAggreg in the YN (Appendix A) and XJ (Appendix A) tiger nut cultivars, respectively. The most stable and the most unstable reference genes in each part were largely the same. The middle rankings of the stability of RGs varied according to the analysis of different algorithms.

### 2.5. Validation of Candidate RGs

To determine the expression stability of potential RGs, an oil body membrane protein, caleosin (CLO), and a type II diacylglycerol acyltransferase (DGAT2-2) were selected to verify the stability of selected RGs by qRT-PCR in various tissues and developmental stages of tiger nut tubers. According to the abovementioned analysis, the two most stable RGs (*UCE2* and *UBL5*) and the most unstable RG (*Rubisco*) were used to normalize the relative expression of the two target genes in the two cultivars of tiger nut (Figure 4 and Figure 5).

CLO is a kind of oil membrane protein that binds to the surface of the oil body in the form of calcium binding protein; the main biological functions of this protein include promoting the degradation of the oil body, maintaining the stability of the oil body, resisting biological stress and participating in the regulation of plant flowering [41,42]. Therefore, the expression of *CLO* is related to the accumulation of oil. As Figure 4A,B shows, similar relative expression levels of *CLO* were obtained using single RGs or a combination of RGs (*UCE2* and *UBL5*) to normalize its expression. The expression patterns of *CLO* differed significantly when using the least stable RG (*Rubisco*) for normalization (Figure 4A,B). The transcripts of *CLO* aggregated notably in three developmental stages of tubers, with the highest level being observed in the tuber swelling stage. However, the expression levels of *CLO* were very low in other non-tuber organs, and the expression levels of underground tissues (including tuber, rhizome, and root) were higher than those of aboveground tissues (young and mature leaf, leaf sheath, and stem apex). The expression levels of *CLO* in the two tiger nut cultivars revealed by transcriptome analysis, shown as FPKM (fragments per kilobase of exon model per million mapped reads) in Figure 4C, were similar to the profiles characterized by qRT-PCR (Figure 4A,B). Comparing the two cultivars, the expression patterns were uniform, and only a slight difference in expression levels was observed between them. These results showed that using different RGs led to different expression patterns of target genes.

DGAT2-2 is an enzyme catalyzing the limited step in TAG biosynthesis [43]. We also analyzed its expression patterns in two cultivars of tiger nut (Figure 5). Generally, there was an approximately 2.5-fold change in *DGAT2-2* expression levels among different tissues in YN tiger nut and a fivefold change in *DGAT2-2* expression levels in XJ tiger nut. The similar expression pattern of *DGAT2-2* between the two cultivars is reflected in Figure 5: On the one hand, tubers in the swelling stage exhibited the highest expression levels of *DGAT2-2* followed by the sheath; on the other hand, mature leaves and shoot apexes showed the lowest transcript levels. The distinctive expression between these two cultivars was mainly observed in mature tubers because mature tubers of YN accumulated more *DGAT2-2* transcripts than those of XJ. Comparing the results of different RGs showed that the relative expression of *DGAT2-2* displayed significant variation when using the most unstable RG (*Rubisco*) for normalization, while similar expression patterns of *DGAT2-2* were presented using single or a combination of RGs (*UCE2* and *UBL5*), as shown in Figure 5A,B. In the transcriptome data, the transcript level of *DGAT2-2* was also higher in leaf sheathes and tubers in the swelling and mature stages (Figure 5C). These results demonstrate that rigorous selection of reliable RGs is crucial to normalize gene expression. If the stability of RGs is poor, the result of normalization will deviate from the real value.

## 3. Discussion

With the in-depth study of gene functions in various species, the utilization of qRT-PCR to detect gene expression levels has become more extensive [25]. The expression levels of appropriate RGs should be constant under different test conditions and in different tissues and should not be affected by any external or internal factors [44].

Tiger nut is a new type of oil crop; this unique plant accumulates oil efficiently in tubers, which are a kind of nutrient organ [20]. To explore the special biological processes of this plant, such as oil accumulation and carbon distribution, it is urgently important to choose the most suitable RGs for transcript normalization. Tubers, the underground storage organ of tiger nuts, are produced at the expense of the rhizome apical meristem [17]. Considering the special underground organs of tiger nut, we not only selected the aboveground tissues (young leaves, mature leaves, leaf sheaths, and stem apexes) but also selected all underground tissues (roots, rhizomes, and tubers of different developmental stages) (Appendix A). Unfortunately, tiger nut plants cultivated in the Yunnan area did not produce flowers or fruit during the growth period; thus, reproductive organs were not available. For further improvement of the preciseness of this study, two cultivars of tiger nut with different tuber traits (smaller oval size (Appendix A) and larger spherical size (Appendix A)) were selected, planted, sampled, and analyzed and named “YN” and “XJ”, respectively.

To compare the stability of candidate RGs more scientifically, three software tools (geNorm, NormFinder, and BestKeeper) were used to analyze the expression stability of 14 candidate RGs. According to the different calculation rules of the three software programs, the stability rankings of candidate genes were generally close and only slightly different. In the consensus rank lists by RankAggreg (Table 5), the most stable and unstable RGs were roughly consistent in the same group of two cultivars: the most stable RGs were generally *UCE2* and *UBL5*, and the least stable RGs were generally *Rubisco* and *PGK*.

The stabilities of *UCE2*, *MDH*, and *UBL5* in the four groups of YN tiger nuts were the top three, while *Rubisco* and *PGK* maintained the lowest stabilities. In addition, *TUB4* showed favorable stability in tuber tissues of YN. Among the aboveground tissues of XJ tiger nut, the most stable candidates were *UCE2*, *EF1α*, and *CYC*, while the most unstable were *Rubisco* and *TUB4*. For underground tissues of XJ, the most stable genes were *UCE2*, *UBL5*, and *TUB4*, whereas the lowest stability was still observed for *Rubisco* and *GAPDH*. Among the tubers at different developmental stages, *TUB4*, *PGK*, and *UCE2* showed the best stability, while *Rubisco* and *18S* showed the lowest stability. For the total organs of XJ, *UCE2*, *UBL5*, and *ADF7* were expressed with the best stability, while *Rubisco* and *PGK* still exhibited the lowest stability. Combining the rankings of the two cultivars, we found that *UCE2* and *Rubisco* maintained the best and worst stability among non-tuber organs, respectively. Moreover, for the two cultivars, *TUB4* exhibited great variability in different types of organs because its stability was the strongest in tuber tissues but the worst after *Rubisco* and *PGK* in the aboveground tissues.

These results show that a combination of *UCE2* and *UBL5* is an appropriate RG for normalizing gene expression in aboveground, underground and total tissues of tiger nuts according to the optimal numbers recommended by geNorm. *UCE* encodes a ubiquitin-conjugating enzyme domain-containing protein that catalyzes ubiquitin transfer to the substrate or E3 ligase and is also a key enzyme in ubiquitin modification of target proteins [45]. In prior studies, *UCE2* showed unstable expression in pepper (Wan et al., 2011), tung trees during seed development [46], and pitaya [47]. However, in keeping with our results, *UCE* has been reported frequently as the most stable reference gene not only in several herbaceous plants, including *Brachypodium distachyon* [48], tobacco [49], red clover [50], *Lycoris aurea* [51], and goosegrass under quizalofop-p-ethyl (a kind of low-toxicity herbicide) stress [52], but also in woody plants, such as poplar [53], *Platycladus orientalis* [54], sweet osmanthus [55], *Plukenetia volubilis* [56], olive trees [57], and rubber trees [58]. As another suitable RG, *UBL5* is a new type of ubiquitin-like protein discovered in the human iris [59] with multiple biological functions in eukaryotes, such as involvement in mRNA precursor splicing [60,61], regulation of the cell cycle [62], regulation of energy metabolism [63], and stress resistance [64]. *UBL* had relatively stable expression in *Plukenetia volubilis* [56], *Jatropha curcas* [27], and European beech [65]; nevertheless, it had the lowest expression level in *Seashore Paspalum* [66].

In addition, for tiger nut tuber tissues, *TUB4* and *UCE2* are a pair of recommended RGs. Tubulin, encoded by *TUB* [67,68], is the main component of plant microtubules [69], which can maintain cell morphology [70], promote intracellular transport [71], and participate in cell elongation [72], movement, and cell division [73,74,75].

According to the analysis of qRT-PCR results, it has been widely recognized that the combination of multiple RGs can provide more accurate and reliable expression profiles than can a single RG [76]. Based on validation of RGs, when *UCE2* or *UBL5* was employed alone or in combination as RGs, the expression patterns of the target genes in different tissues were largely consistent. It demonstrated that the expression patterns of the target genes were almost the same as with one or two RGs. However, when *Rubisco* was used as an RG to normalize target genes, the patterns of *CLO* and *DGAT2-2* expression were clearly inconsistent with that of the pattern obtained with the two optimum RGs, especially in mature leaves and all underground tissues, suggesting the most unstable characteristic of *Rubisco*.

## 4. Materials and Methods

### 4.1. Plant Material

Two cultivars of tiger nut were collected from Shihezi University in Xinjiang (XJ), Northwest China and Heshun County in Yunnan (YN), Southwest China. XJ tiger nut produces larger average spherical tubers (Appendix A), while YN tiger nut produces smaller oval tubers (Appendix A). On March 15, 2018, tubers were sown in a field at the Xishuangbanna Tropical Botanical Garden (XTBG, 21° 54′ N, 101° 46′ E, 580 m in altitude) of the Chinese Academy of Sciences in Mengla County, Yunnan Province, Southwest China. Six tissues (young leaf, mature leaf, leaf sheath, root, rhizome, stem apex) were collected on 4 May 2018, and tubers of three developmental stages were collected at 40, 80, and 120 days after sowing (DAS), which are the stages of tuber formation, swelling and maturity in turn. All samples were harvested, washed and surface dried and then frozen in liquid nitrogen and immediately stored at −80 °C until required for further analysis. In total, three biological replicates were collected, and each biological replicate contained tissue sampled from ten individual plants.

### 4.2. Selection of Candidate RGs and Primer Design

Fourteen candidate RGs were selected based on RNA-Seq data from tiger nut tissues [21], including *18S ribosomal RNA* (*18S*), *actin* (*ACT*), *actin-depolymerizing factor 7* (*ADF7*), *elongation factor 1-alpha* (*EF1α*), *elongation factor 2* (*EF2*), *glyceraldehyde-3-phosphate dehydrogenase* (*GAPDH*), *ribosomal protein L11* (*RPL11*), *ribulose bisphosphate carboxylase* (*Rubisco*), *ubiquitin-conjugating enzyme 2* (*UCE2*), *ubiquitin-like protein 5* (*UBL5*), *cyclophilin* (*CYC*), *phosphoglycerate kinase* (*PGK*), *malate dehydrogenase* (*MDH*), and *tubulin beta-4* (*TUB4*). The cDNA sequences of candidate RGs and two genes for validation were obtained from Yang et al. [21] (Appendix A). Primers used for qRT-PCR analyses (Table 1) were designed using Primer Premier 5.0 with the following parameters: melting temperature (T_m_) values ranging from 50 to 60 °C, GC percent of 40–60%, primer lengths of 18–22 bp and product length of 100–300 bp. Tenfold serial dilutions of cDNA were used to determine the slope of the standard curve to calculate the amplification efficiency of primers (*E* = 10(−1/slope of the standard curve)).

### 4.3. RNA Isolation, cDNA Synthesis, and qRT-PCR

Total RNA was extracted by silica particles [77]. Plant tissues (0.2–0.5 g) frozen in liquid nitrogen were ground into powder rapidly and then mixed with 400 μL of RNA extraction buffer (5 M NaCl and 100 mM Tris–HCl [pH = 8.0]), 100 μL of 2-mercaptoethanol, and 300 μL of water-saturated phenol in a 2-mL centrifuge tube. After vortexing and incubating at room temperature (RT) for 5 min, 300 μL of chloroform/isoamyl alcohol (24:1), and 200 μL of 3 M NaAc (pH = 4.0) were added to the mixture. After vortexing and centrifuging (15,000× *g*, 10 min, 4 °C), the upper aqueous phase was carefully pipetted into a 1.5-mL centrifuge tube, then 200 μL of absolute ethanol and 25 μL of silica suspension (1 g/mL) were added and mixed. After centrifuging (15,000× *g*, 30 s, RT), the liquid was poured out, and the silica pellet was rinsed twice with 75% ethanol (prepared with DEPC water). The silica pellet dried in vacuum was suspended with 50 μL of DEPC water and centrifuged (15,000× *g*, 10 min, RT). The supernatant containing the total RNA was transfer to a new centrifuge tube and stored at −80 °C. RNA quality was determined by 2.0% agarose gel electrophoresis. The concentration and purity of total RNA were determined using a NanoDrop 2000c Spectrophotometer (Thermo Scientific, Wilmington, DE, USA). The ratio of OD_260_/OD_280_ of total RNA between 1.90 and 2.10 was considered to meet the required quality for further experiments.

Two µg of total RNA from each sample was reverse-transcribed into cDNAs using the PrimeScript^®^ RT Reagent Kit with gDNA Eraser (TaKaRa, Dalian, China). The reverse transcriptional reaction included 4 μL of 5× PrimerScript^®^ Buffer 2, 1 μL of PrimerScript^®^ RT-Enzyme Mix I, 1 μL of RT Primer Mix, 10 μL of the reaction liquid of genomic DNA removal, and RNase-Free ddH_2_O up to 20 μL, reacting at 37 °C for 15 min and then terminating at 85 °C for 5 s. qRT-PCR was performed using LightCycler^®^ 480 SYBR Green I Master Mix with the Roche 480 real-time PCR detection system according to the manufacturer’s instructions (Roche Diagnostics, Indianapolis, IN, USA). One µL of cDNA was added to each qRT-PCR reaction mix (20 µL), containing 0.25 µM of each primer and 10 µL of SYBR^®^ Premix Ex Taq™ II. Amplifications were performed with the following program: 95 °C for 3 s; 40 cycles of 95 °C for 10 s, 60 °C for 30 s, 72 °C for 3 s. Melting curve analysis was performed as follows: 95 °C for 15 s, 60 °C increased to 95 °C with temperature increment of 0.6 °C/s, cooling at 16 °C.

### 4.4. Data Analysis

Several software tools were used to evaluate the stability of the 14 selected candidate RGs across all the experimental sets, including geNorm [26], NormFinder [37], and BestKeeper [38]. The expression levels of the candidate RGs were determined by cycle threshold (Ct) values. All the procedures of statistical analyses by the above packages were conducted according to the manufacturer’s instructions.

GeNorm software is specifically used to screen RGs and determine the optimal number of RGs in real-time quantitative PCR. The result of the analysis is that two or more internal RG combinations were selected to correct the data, which makes the relative quantitative results more accurate. The geNorm program selects the optimal RG with stability by calculating the *M* value of each RG. The criterion is that the smaller the M value is, the better the stability of the RG is; otherwise, the stability is worse. This software can also calculate the paired variation *V* value of the normalized factor after the introduction of a new internal RG and determine the number of optimal internal RGs based on the Vn/Vn + 1 value. The default *V* value is 0.15. If the value of Vn/Vn + 1 is less than 0.15, the number of optimal internal RGs is n; if the value is more than 0.15, the number of suitable internal RGs is *n* + 1.

NormFinder is another program for screening stable internal RGs [37]. The calculation principle of this program is similar to that of the geNorm program. It also obtains the stable value of the internal RG and then selects the value based on the stable value. The most suitable internal RG is determined, and the criterion is that the internal RG with the lowest expression stability value is the most suitable internal RG. The NormFinder program can not only compare the expression differences of candidate internal RGs but also calculate the variation between sample groups, but the program can only screen out one of the most suitable internal RGs.

BestKeeper is a program for analysing the expression of internal RGs [38]. The correlation coefficient (*r*), standard deviation (SD), and coefficient of variation (CV) that can generate pairings between each gene can be obtained through the calculation of the program. Next, by comparing the values of each value, a stable internal RG is finally determined. The determination principle is that the larger the correlation coefficient, the smaller the standard deviation and the coefficient of variation, the better the stability of the RG, and conversely, the worse the stability; when SD > 1, the expression of the RG is unstable.

### 4.5. Statistical Method for Rank Aggregation

The RankAggreg package of R software was used to combine the stability measurements obtained from the four methods and establish a consensus rank of RGs. To generate a consensus from the data produced by geNorm, BestKeeper, and NormFinder, we aggregated the obtained ranking lists by applying the RankAggreg (ver. 0.6.5) package of R software as done previously. RankAggreg is a package that provides algorithms able to combine different ranking lists [39].

### 4.6. Validation of the Candidate RGs

To examine the expression stability of potential RGs, the relative expression levels of two lipid-related genes (*CLO* and *DGAT2-2*) were analyzed in various tissues in two cultivars of tiger nut. CLO is a kind of oil-body-related protein [78], and DGAT2-2 is a type II diacylglycerol acyltransferase [79]. The relative expression levels of these two genes were normalized separately to the most stable and least stable RGs analyzed by the mentioned four algorithms. The qRT-PCR amplification conditions of the genes were the same as those described above. The relative expression levels of target genes were calculated according to the 2^−ΔΔCt^ method and presented as fold change.

### 4.7. Library Construction, Sequencing, and Assembly of Transcriptomes

The total RNAs from 54 samples (nine tissues of each two cultivars, three replicates per tissue) described above were isolated for construction of transcriptome libraries. The quality of the libraries was assessed using an Agilent 2100 Bioanalyzer. Transcriptome sequencing were performed on the Illumina NovaSeq 6000 platform by Novogene Bioinformatics Technology (Tianjin, China) to generate 150-nucleotide-long paired-end sequence reads. Clean reads were obtained from the raw reads by removing adapter sequences and low-quality reads. De novo transcriptome assembly was performed using Trinity (v2.4.0) with default parameters [80]. The clean reads were matched to the *Cyperus esculentus* unigenes using RSEM software [81]. The frequency of individual reads was normalized to FPKM [82]. RNA-Seq data reported in this study were deposited at NCBI under the BioProject ID PRJNA703731.

## 5. Conclusions

In this study, our purpose was to select and validate the optimal RGs for normalization of qRT-PCR analysis in tiger nut to establish a foundation for further research on its metabolic pathway and regulatory network. The expression stabilities of 14 potential RGs were evaluated in various tissues from two cultivars of tiger nut, including three leaf-related samples, stem apexes, roots, rhizomes, and tubers of three developmental stages. In the evaluation stage, three kinds of analysis software were utilized for ranking first, and then RankAggreg was used to integrate multiple rankings into a consensus list. *UCE2*, *MDH* and *UBL5* were the top three stable RGs across all tissues of YN tiger nut. *UCE2*, *UBL5* and *ADF7* were the most stable candidates across all tissues of XJ tiger nut. According to the rank list in the “Tissues of two cultivars” column of Table 5, a combination of *UCE2* and *UBL5* was the best RG for all tissues of the two cultivars. Similarly, a combination of *TUB4* and *UCE2* was the most suitable for tuber tissues of the two cultivars. Transcriptome data can be used to select candidate RGs and analyze the expression levels of two target genes. These results provide a basis and convenient resource for future research performing gene expression analysis in tiger nut.

## Figures and Tables

**Figure 1 ijms-22-02569-f001:**
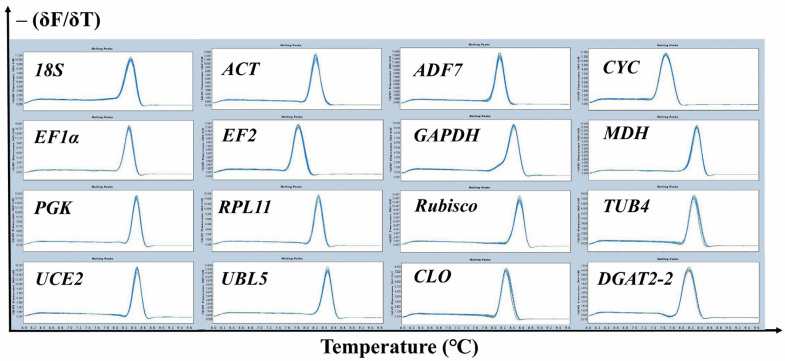
Melt curve analysis of 14 candidate reference genes and two target genes assessed across all tissues of two cultivars, “YN” and “XJ”, of tiger nut (including young leaves, mature leaves, leaf sheathes, stem apexes, roots, rhizomes and tubers in three developmental stages) and showed a single peak for each primer pair at a specific annealing temperature. The –(δF/δT) value represents the raw fluorescence (F) versus temperature (T) values.

**Figure 2 ijms-22-02569-f002:**
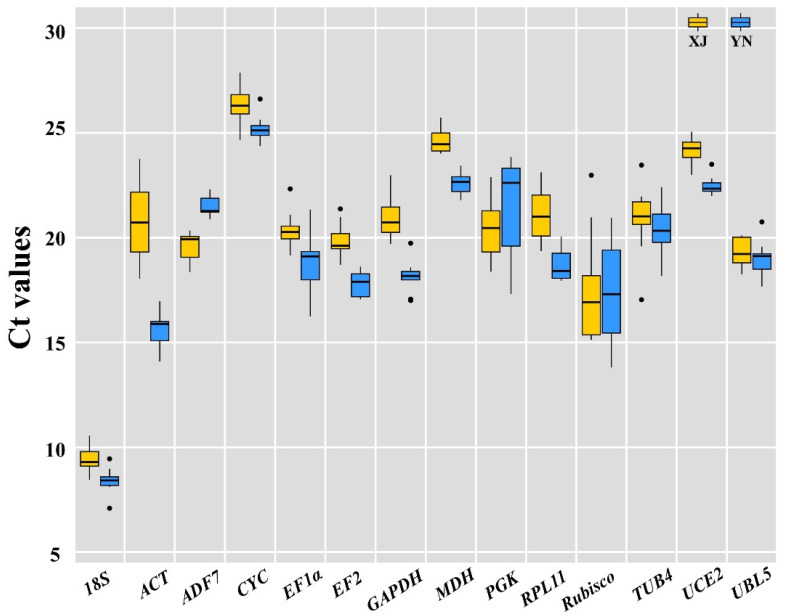
Distribution of threshold cycle (C_t_) values of 14 candidate reference genes across all 18 samples. Yellow boxes represent XJ tiger nuts, and blue boxes represent YN tiger nuts. The box indicates the 25th and 75th percentiles. The line across the box indicates the median, and the cross depicts the mean.

**Figure 3 ijms-22-02569-f003:**
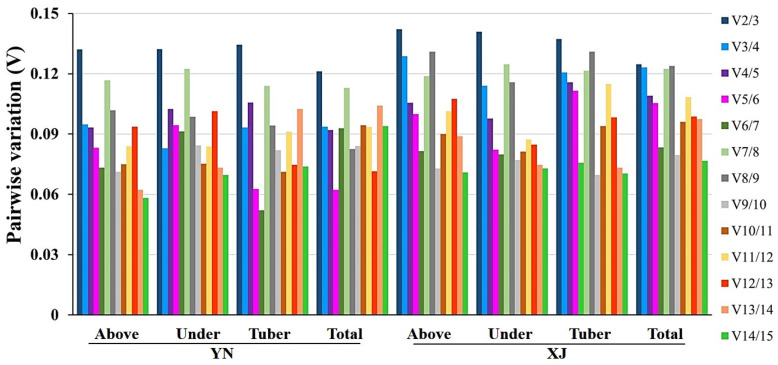
Pairwise variation (V) analyses of 14 candidate reference genes to use for data normalization in two cultivars, “YN” and “XJ”, of tiger nut, as computed by geNorm. This chart contains several tissue groups, including above (aboveground), under (underground), tubers, and total samples of two cultivars.

**Figure 4 ijms-22-02569-f004:**
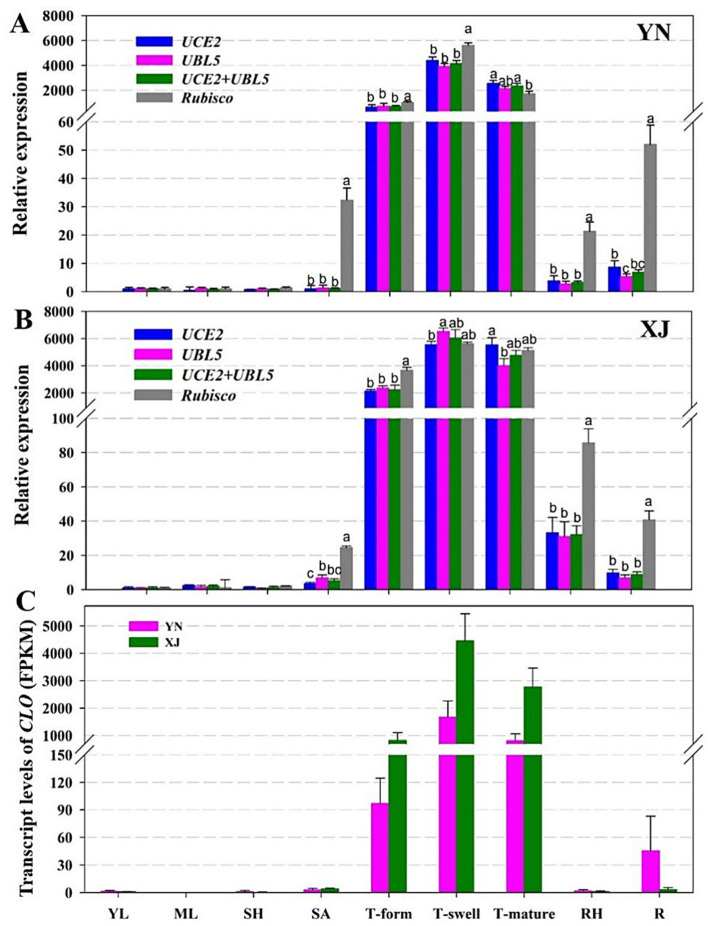
Expression profiles of *CLO* in different tissues of YN and XJ tiger nuts. (**A**) Relative expressions in YN tissues by qRT-PCR. (**B**) Relative expressions in XJ tissues by qRT-PCR. *UCE2*, *UBL5*, and *UCE2* + *UBL5* were used as the one or two most stable reference genes. *Rubisco* was used as the least stable reference gene, which is shown in grey bars. (**C**) Transcript levels of *CLO*. The horizontal axis shows nine tissues of tiger nut (YL: young leaves, ML: mature leaves, SH: sheath, SA: shoot apexes, T-form: tubers in the formation stage, T-swell: tubers in the swelling stage, T-mature: tubers in the mature stage, RH: rhizome, R: root). Data are represented as the mean ± SD. Bars with different letters (a, b, and c) are significantly different from each other in the expression of the target gene based on three biological replications (*p* < 0.05, *t* test; *n* = 3).

**Figure 5 ijms-22-02569-f005:**
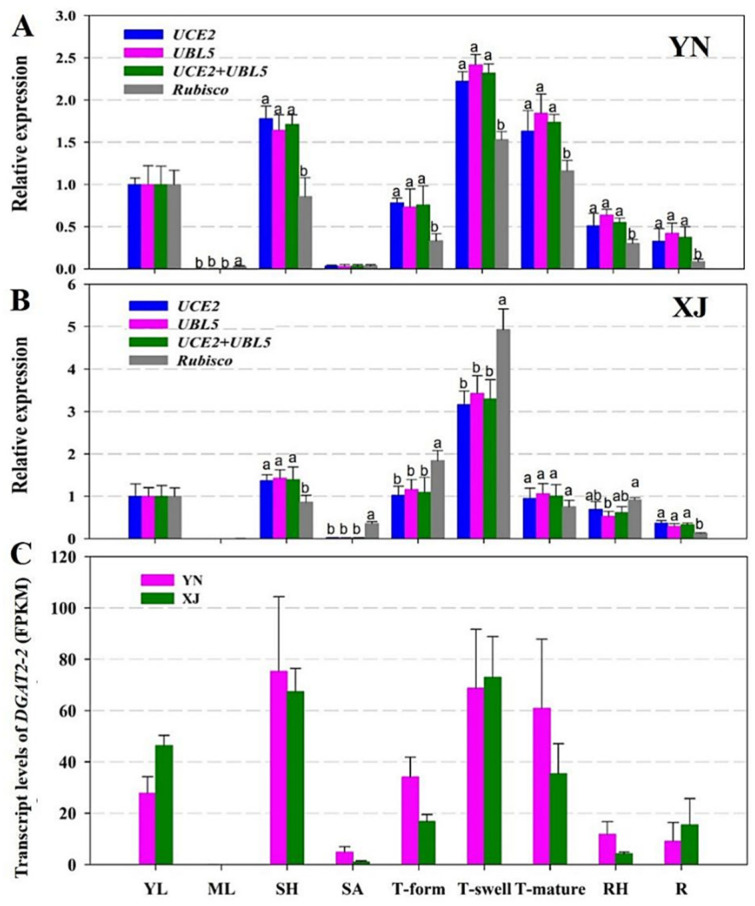
Expression profiles of *DGAT2-2* in different tissues of YN and XJ tiger nut. (**A**) Relative expressions in YN tissues by qRT-PCR. (**B**) Relative expressions in XJ tissues by qRT-PCR. *UCE2*, *UBL5*, and *UCE2* + *UBL5* were used as the one or two most stable reference genes. *Rubisco* was used as the least stable reference gene, which is shown in grey bars. (**C**) Transcript levels of *DGAT2-2*. The horizontal axis shows nine tissues of tiger nut (YL: young leaves, ML: mature leaves, SH: sheath, SA: shoot apexes, T-form: tubers in the formation stage, T-swell: tubers in the swelling stage, T-mature: tubers in the mature stage, RH: rhizome, R: root). Data are represented as the mean ± SD. Bars with different letters (a, b, and c) are significantly different from each other in the expression of the target gene based on three biological replications (*p* < 0.05, *t* test; *n* = 3).

**Table 1 ijms-22-02569-t001:** Primers used for qRT-PCR analyses in this study. Potential reference genes were the first fourteen, and the last two genes were used to validate reference genes. The details of gene names, description, unigene ID, primer sequence, amplicon length, primers Tm (°C), amplification efficiency (E (%)), and correlation coefficient (R^2^) are presented.

Gene	Description	Primer Sequence (5′-3′; F, forward; R, Reverse)	Amplicon Length (bp)	Primers Tm (°C)	E (%)	R^2^
*18S*	18S Ribosomal RNA	F: GGAAGTTTGAGGCAATAACAGG	140	53.71/55.18	97.29	0.9931
		R: TATCCCCATCACGATGAAATTTCTC				
*ACT*	Actin	F: CTCAACCCCAAGGCCAACA	146	53.52/53.81	115.7	0.9923
		R: CCATCACCAGAGTCAAGAACAATA				
*ADF7*	Actin-depolymerizing factor 7	F: GACACCGCAAGGGTAAGG	118	52.84/52.25	103.59	0.9942
		R: CAAGCCCCATCTCAGTAGG				
*CYC*	Cyclophilin	F: GGTGAAAAGGGTATCGGTC	140	51.73/52.41	105.16	0.9998
		R: TTTCCGTAGATGGACTCGC				
*EF1α*	Elongation factor	F: CTGGTATGCTTGTGACATTTGG	175	54.07/53.96	98.64	0.9931
	1-alpha	R: TCGTCCTTGGAGTTGGAGG				
*EF2*	Elongation factor2	F: TGTCCTTCCGTGAGACCGTA	203	55.85/53.17	103.93	0.9959
		R: TCCTTGTCCCATCCGAACT				
*GAPDH*	Glyceraldehyde-3-phosphate dehydrogenase	F: ATTCCCAGCAGCACTGGTG	93	52.75/51.93	110.61	0.9962
		R: AGTTGGCACACGGAAAGCC				
*MDH*	Malate	F: ACCCTCTTGTGTCGGTTCTT	189	54.44/52.28	95.78	0.9944
	Dehydrogenase	R: TTGTCATGCCTGGTTTACG				
*PGK*	Phosphoglycerate	F: AGAAACCAAGGCTTCGTCA	168	52.24/52.98	92.54	0.992
	Kinase	R: AAGGGAGTCACAACCATCATT				
*RPL11*	Ribosomal protein	F: CTGGATGCTTTGGATTCGG	174	51.82/53.5	96.39	0.9965
	L11	R: CCTTGGTAACTCTGTGCTGGA				
*Rubisco*	Ribulose bisphosphate carboxylase	F: ATGTCTACGTGGTGGACTTGAT	124	52.23/51.92	108.65	0.9973
		R: TGTTTCGGCTTGTGCTTTAT				
*TUB4*	Tubulin beta-4	F: CAGGAAGGAGGCTGAAAAT	156	52.06/53.6	118.22	0.9931
		R: GAGGGGAAGACAGAGAAGGT				
*UCE2*	Ubiquitin-conjugating enzyme 2	F: ATCATCAAGGAGACCCAGCG	183	55.14/53.39	98.21	0.9998
		R: CTTAGGGGCAGCCATAGGA				
*UBL5*	Ubiquitin-like	F: ATAATCCCCGTATTTCCACTGC	96	54.34/55.54	109.54	0.9952
	protein 5	R: GAATCTATCCTATCCACGCTCTCT				
*CLO*	Caleosin	F: ACGGCATTGTTTATCCCTGG	178	53.86/54.32	117.02	0.9914
		R: TGTTTGGCTCTGTGTATGTTGTGT				
*DGAT2-2*	Diacylglycerol	F: CAGGTGGTGTTCAAGAGATGCT	126	54.85/53.26	102.45	0.9953
	O-acyltransferase 2-2	R: CAAAGGAGAAAACAGGGACAAGT				

**Table 2 ijms-22-02569-t002:** Expression stability ranking of candidate reference genes in different tissues was calculated using geNorm. The average expression stability (M value) is listed, and the stability decreased from top to bottom.

geNorm Rank	Tissues of YN								Tissues of XJ							
	Aboveground		Underground		Tuber		Total		Aboveground		Underground		Tuber		Total	
	Gene	*M*	Gene	*M*	Gene	*M*	Gene	*M*	Gene	*M*	Gene	*M*	Gene	*M*	Gene	*M*
1	*UCE2*	0.316	*UCE2*	0.048	*UCE2*	0.041	*UCE2*	0.403	*UCE2*	0.292	*UCE2*	0.215	*TUB4*	0.220	*UCE2*	0.267
2	*MDH*	0.320	*TUB4*	0.062	*TUB4*	0.053	*MDH*	0.409	*UBL5*	0.292	*TUB4*	0.233	*UBL5*	0.241	*ADF7*	0.296
3	*UBL5*	0.398	*ACT*	0.120	*ACT*	0.115	*TUB4*	0.422	*EF1α*	0.303	*ACT*	0.270	*UCE2*	0.529	*UBL5*	0.382
4	*ADF7*	0.414	*EF2*	0.380	*EF2*	0.275	*UBL5*	0.593	*CYC*	0.341	*UBL5*	0.285	*PGK*	0.589	*18S*	0.474
5	*ACT*	0.463	*MDH*	0.514	*MDH*	0.373	*CYC*	0.665	*GAPDH*	0.345	*ADF7*	0.310	*ADF7*	0.591	*CYC*	0.510
6	*CYC*	0.560	*RPL11*	0.598	*GAPDH*	0.461	*EF2*	0.722	*ACT*	0.346	*18S*	0.348	*CYC*	0.591	*EF2*	0.534
7	*TUB4*	0.591	*CYC*	0.657	*CYC*	0.542	*RPL11*	0.756	*MDH*	0.367	*EF1α*	0.375	*RPL11*	0.679	*GAPDH*	0.551
8	*EF2*	0.593	*EF1α*	0.724	*RPL11*	0.608	*GAPDH*	0.818	*ADF7*	0.382	*PGK*	0.386	*MDH*	0.711	*TUB4*	0.574
9	*EF1α*	0.623	*GAPDH*	0.775	*Rubisco*	0.655	*18S*	0.869	*TUB4*	0.384	*RPL11*	0.397	*EF1α*	0.770	*EF1α*	0.576
10	*RPL11*	0.641	*UBL5*	0.814	*EF1α*	0.684	*ACT*	0.969	*18S*	0.394	*CYC*	0.419	*EF2*	0.831	*MDH*	0.612
11	*GAPDH*	0.654	*18S*	0.878	*UBL5*	0.727	*ADF7*	1.014	*EF2*	0.446	*EF2*	0.438	*ACT*	0.855	*RPL11*	0.643
12	*18S*	0.740	*PGK*	0.940	*18S*	0.791	*EF1α*	1.089	*RPL11*	0.485	*MDH*	0.448	*18S*	0.975	*PGK*	0.777
13	*PGK*	*1.163*	*ADF7*	*0.995*	*PGK*	*0.865*	*PGK*	*1.365*	*PGK*	*0.513*	*GAPDH*	*0.485*	*GAPDH*	*1.320*	*ACT*	*0.790*
14	*Rubisco*	*1.304*	*Rubisco*	*1.182*	*ADF7*	*0.954*	*Rubisco*	*1.552*	*Rubisco*	*0.956*	*Rubisco*	*0.660*	*Rubisco*	*1.409*	*Rubisco*	*0.963*

The aboveground part including young leaf, mature leaf, leaf sheath, and stem apex. The underground part including root, rhizome, and all tubers. The tuber part contains three developmental stages of tubers collected at 40, 80, and 120 days after sowing (DAS), which are the stages of tuber formation, growth, and maturity, respectively. The total represents all samples of each cultivar.

**Table 3 ijms-22-02569-t003:** Stability “*M*” value of candidate reference genes was determined by NormFinder. The stability decreased from top to bottom.

Norm- Finder Rank	Tissues of YN								Tissues of XJ							
	Aboveground		Underground		Tuber		Total		Aboveground		Underground		Tuber		Total	
	Gene	*M*	Gene	*M*	Gene	*M*	Gene	*M*	Gene	*M*	Gene	*M*	Gene	*M*	Gene	*M*
1	*UCE2*	0.152	*TUB4*	0.211	*TUB4*	0.221	*UCE2*	0.359	*UCE2*	0.233	*TUB4*	0.190	*TUB4*	0.191	*UBL5*	0.231
2	*UBL5*	0.320	*RPL11*	0.290	*RPL11*	0.236	*UBL5*	0.409	*EF1α*	0.292	*UBL5*	0.233	*PGK*	0.241	*UCE2*	0.345
3	*MDH*	0.225	*MDH*	0.335	*UCE2*	0.264	*EF2*	0.494	*CYC*	0.475	*UCE2*	0.409	*ADF7*	0.772	*ADF7*	0.358
4	*ADF7*	0.227	*UCE2*	0.438	*MDH*	0.308	*MDH*	0.545	*PGK*	0.500	*CYC*	0.476	*CYC*	0.837	*CYC*	0.413
5	*EF1α*	0.444	*ACT*	0.454	*EF2*	0.324	*ACT*	0.557	*ACT*	0.550	*EF1α*	0.493	*UBL5*	0.862	*EF2*	0.430
6	*ACT*	0.450	*EF2*	0.458	*EF1α*	0.350	*ADF7*	0.592	*MDH*	0.606	*RPL11*	0.549	*UCE2*	0.865	*18S*	0.442
7	*GAPDH*	0.591	*GAPDH*	0.473	*GAPDH*	0.355	*GAPDH*	0.606	*RPL11*	0.632	*18S*	0.598	*RPL11*	0.884	*ACT*	0.455
8	*CYC*	0.635	*CYC*	0.476	*ACT*	0.473	*CYC*	0.627	*TUB4*	0.636	*ACT*	0.604	*MDH*	1.003	*GAPDH*	0.479
9	*EF2*	0.639	*EF1α*	0.531	*UBL5*	0.479	*EF1α*	0.643	*EF2*	0.667	*EF2*	0.646	*EF1α*	1.003	*EF1α*	0.485
10	*TUB4*	0.691	*UBL5*	0.563	*CYC*	0.496	*TUB4*	0.698	*GAPDH*	0.724	*MDH*	0.773	*ACT*	1.037	*TUB4*	0.714
11	*RPL11*	0.694	*PGK*	0.679	*Rubisco*	0.527	*RPL11*	0.722	*UBL5*	0.817	*PGK*	0.787	*EF2*	1.154	*MDH*	0.951
12	*18S*	0.833	*18S*	0.725	*PGK*	0.753	*18S*	0.754	*18S*	0.982	*GAPDH*	1.181	*18S*	1.163	*RPL11*	1.054
13	*PGK*	1.419	*ADF7*	0.800	*18S*	0.807	*PGK*	1.753	*Rubisco*	1.327	*ADF7*	1.194	*GAPDH*	1.260	*Rubisco*	1.277
14	*Rubisco*	1.702	*Rubisco*	1.316	*ADF7*	0.989	*Rubisco*	1.936	*ADF7*	1.619	*Rubisco*	1.818	*Rubisco*	2.045	*PGK*	1.494

**Table 4 ijms-22-02569-t004:** Expression stability of 14 candidate reference genes in different tissues was determined by BestKeeper. The standard deviation [SD] and coefficient of variation [CV] were given (the lower, the more stable). The stability decreased from top to bottom.

Best- Keeper Rank	Tissues of YN												Tissues of XJ											
	Aboveground			Underground			Tuber			Total			Aboveground			Underground			Tuber			Total		
	Gene	SD	CV	Gene	SD	CV	Gene	SD	CV	Gene	SD	CV	Gene	SD	CV	Gene	SD	CV	Gene	SD	CV	Gene	SD	CV
1	*MDH*	0.27	1.05	*UCE2*	0.42	1.35	*TUB4*	0.10	1.16	*UCE2*	0.44	1.94	*MDH*	0.09	0.96	*UCE2*	0.55	1.31	*UCE2*	0.23	1.13	*UCE2*	0.57	1.31
2	*CYC*	0.30	1.36	*MDH*	0.43	1.93	*UCE2*	0.28	1.54	*MDH*	0.48	2.23	*EF2*	0.23	0.96	*MDH*	0.63	1.52	*TUB4*	0.29	1.24	*18S*	0.65	1.70
3	*UCE2*	0.39	1.85	*ACT*	0.45	5.27	*EF2*	0.40	1.80	*EF2*	0.50	2.23	*UCE2*	0.27	1.37	*UBL5*	0.78	3.04	*PGK*	0.45	2.12	*UBL5*	0.67	2.06
4	*ADF7*	0.42	1.82	*TUB4*	0.48	2.20	*MDH*	0.41	1.92	*18S*	0.55	3.07	*GAPDH*	0.31	1.28	*TUB4*	0.84	3.02	*ACT*	0.51	2.37	*ADF7*	0.68	2.55
5	*UBL5*	0.60	3.28	*ADF7*	0.49	2.16	*18S*	0.52	2.29	*UBL5*	0.61	2.23	*CYC*	0.36	1.72	*18S*	0.84	3.23	*MDH*	0.53	2.86	*EF2*	0.72	2.70
6	*EF2*	0.60	3.15	*EF2*	0.56	3.10	*GAPDH*	0.55	2.38	*CYC*	0.64	2.52	*18S*	0.36	1.38	*ADF7*	0.88	3.54	*CYC*	0.55	2.20	*EF1α*	0.83	3.20
7	*18S*	0.66	3.73	*GAPDH*	0.62	2.70	*CYC*	0.59	3.29	*RPL11*	0.74	3.96	*ADF7*	0.40	2.04	*EF1α*	0.92	3.44	*RPL11*	0.56	2.19	*MDH*	0.85	3.17
8	*RPL11*	0.69	4.46	*PGK*	0.63	3.21	*RPL11*	0.70	3.56	*GAPDH*	0.77	4.24	*EF1α*	0.52	2.61	*ACT*	0.96	4.00	*GAPDH*	0.62	3.08	*CYC*	0.90	2.42
9	*ACT*	0.78	5.09	*UBL5*	0.67	3.65	*Rubisco*	0.74	2.90	*ADF7*	0.86	4.54	*UBL5*	0.52	2.70	*RPL11*	1.00	3.88	*UBL5*	0.65	4.07	*GAPDH*	0.97	3.60
10	*GAPDH*	0.96	5.26	*RPL11*	0.81	3.22	*EF1α*	0.79	4.57	*ACT*	0.88	5.67	*RPL11*	0.75	3.40	*PGK*	1.07	4.01	*ADF7*	0.80	4.24	*RPL11*	1.22	4.76
11	*EF1α*	1.06	5.99	*CYC*	0.88	5.54	*UBL5*	0.81	4.37	*TUB4*	1.26	6.17	*TUB4*	0.83	4.31	*EF2*	1.09	4.47	*18S*	0.89	4.66	*PGK*	1.46	6.17
12	*TUB4*	1.10	5.59	*18S*	0.98	4.96	*ACT*	0.92	4.55	*EF1α*	1.40	7.43	*PGK*	1.08	4.85	*CYC*	1.14	3.37	*EF1α*	0.90	4.63	*TUB4*	1.69	7.09
13	*Rubisco*	2.31	7.59	*EF1α*	1.05	4.98	*PGK*	0.92	4.37	*Rubisco*	2.34	7.50	*ACT*	2.35	5.72	*GAPDH*	1.25	4.98	*EF2*	1.01	4.83	*ACT*	1.81	7.79
14	*PGK*	2.35	8.03	*Rubisco*	1.58	8.53	*ADF7*	0.96	4.75	*PGK*	2.41	8.22	*Rubisco*	2.35	6.30	*Rubisco*	2.74	6.33	*Rubisco*	1.21	6.15	*Rubisco*	2.60	7.83

**Table 5 ijms-22-02569-t005:** Consensus rank list of 14 reference genes integrated by RankAggreg. It represents comparisons between the two cultivars YN and XJ, as well as among the different tissue groups, including above (aboveground), under (underground), tuber, and total samples. The stability decreased from top to bottom.

Rank	Tissues of YN				Tissues of XJ				Tissues of Two Cultivars			
	Above	Under	Tuber	Total	Above	Under	Tuber	Total	Above	Under	Tuber	Total
1	*UCE2*	*UCE2*	*TUB4*	*UCE2*	*UCE2*	*UCE2*	*TUB4*	*UCE2*	*UCE2*	*UCE2*	*TUB4*	*UCE2*
2	*MDH*	*TUB4*	*UCE2*	*MDH*	*EF1α*	*UBL5*	*PGK*	*UBL5*	*MDH*	*TUB4*	*UCE2*	*UBL5*
3	*UBL5*	*MDH*	*EF2*	*UBL5*	*CYC*	*TUB4*	*UCE2*	*ADF7*	*EF1α*	*ACT*	*MDH*	*EF2*
4	*ADF7*	*ACT*	*MDH*	*EF2*	*MDH*	*ADF7*	*CYC*	*18S*	*UBL5*	*UBL5*	*CYC*	*CYC*
5	*ACT*	*EF2*	*RPL11*	*CYC*	*EF2*	*18S*	*UBL5*	*EF2*	*CYC*	*MDH*	*RPL11*	*MDH*
6	*EF2*	*RPL11*	*GAPDH*	*GAPDH*	*UBL5*	*ACT*	*ADF7*	*CYC*	*EF2*	*RPL11*	*EF2*	*ADF7*
7	*CYC*	*GAPDH*	*ACT*	*TUB4*	*GAPDH*	*EF1α*	*MDH*	*EF1α*	*ACT*	*ADF7*	*PGK*	*18S*
8	*EF1α*	*CYC*	*CYC*	*ACT*	*ACT*	*RPL11*	*ACT*	*GAPDH*	*ADF7*	*18S*	*ACT*	*GAPDH*
9	*TUB4*	*UBL5*	*EF1α*	*RPL11*	*18S*	*MDH*	*RPL11*	*MDH*	*GAPDH*	*EF1α*	*UBL5*	*TUB4*
10	*GAPDH*	*EF1α*	*Rubisco*	*18S*	*PGK*	*CYC*	*EF1α*	*TUB4*	*18S*	*EF2*	*GAPDH*	*EF1α*
11	*RPL11*	*PGK*	*UBL5*	*ADF7*	*RPL11*	*PGK*	*EF2*	*ACT*	*RPL11*	*CYC*	*EF1α*	*ACT*
12	*18S*	*18S*	*18S*	*EF1α*	*ADF7*	*EF2*	*GAPDH*	*RPL11*	*TUB4*	*GAPDH*	*ADF7*	*RPL11*
13	*PGK*	*ADF7*	*PGK*	*PGK*	*TUB4*	*GAPDH*	*18S*	*PGK*	*PGK*	*PGK*	*18S*	*PGK*
14	*Rubisco*	*Rubisco*	*ADF7*	*Rubisco*	*Rubisco*	*Rubisco*	*Rubisco*	*Rubisco*	*Rubisco*	*Rubisco*	*Rubisco*	*Rubisco*

## Data Availability

The transcriptome datasets reported in this study are available at the NCBI under the BioProject ID PRJNA703731.

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
