# Peer review of "Selection and Validation of Reference Genes for qRT-PCR Analysis in the Oil-Rich Tuber Crop Tiger Nut (Cyperus esculentus) Based on Transcriptome Data"

_ijms, 2021, doi:10.3390/ijms22052569_

Round 1

Reviewer 1 Report

The paper aims to identify a set of reference genes (RGs) in the Oil-Rich Tuber Crop Tiger Nut (Cyperus esculentus), a crop that is characterized by a oil-rich tuber. The selection and use of proper RGs could be essential to perform gene expression analysis able to elucidate new genes and pathways related to lipid accumulation. Based on transcriptomic data, the study considered 14 possible RGs across 18 different tissues of two cultivars, and the RGs were analysed using three different algorithms. Additionally, the expression of two genes  caleosin (CLO) and diacylglycerol acyltransferase 2-2 (DGAT2-2) was performed to validate the results. 

In my opinion the research design is appropriate, results are clearly presented (including the supplementary materials) and discussed. 

Author Response

Thank you very much for your positive comments.

Reviewer 2 Report

In this paper, “Selection and validation of reference genes for aRT-PCR analysis in the oil-rich tuber crop tiger nut (Cyperus esculentus) based on transcriptome data”, the authors address a problem inherent when studying RNA-based gene expression in any non-model system, the lack of reference genes. These are essential for standardization of RNA levels across tissues and replicates. Experiments to find and evaluate genes suitable to act as reference genes are not trivial and this reviewer understands why the authors have chosen to present this portion of their studies as an independent manuscript rather than incorporating it into a larger study.

I think the experiments and data presentation are very good and I agree with their interpretations.  Before this study is published I would like to authors to address several items.

  1. Some of the language in the first paragraph of the introduction should be changed to make it more scientific. The phrase “natural cooking oil” on line 40 is too colloquial. I have no idea what “natural cooking oil” is. Aren’t all cooking oils derived from plant or animal sources, which are “natural”.

The phrase “beneficial for human health” on line 50 is also too colloquial. Lipids are an essential component of the human diet and all may be considered “healthy” since they provide calories. The authors should specifically explain what distinguishes oleic acid (C18:1) from other lipids regarding the human diet?  For example, is it less likely to contribute to heart disease? The unqualified terms “natural” and “healthy” might work for advertisers and marketing companies to sell a product to the general public but they don’t belong in a scientific study without further explanation.   

  1. I think the reference to unpublished transcriptome data on line 74 of the introduction belongs somewhere else in the manuscript, perhaps the materials and methods. I also think that, since this data is very important to this study and is the basis of the selection of the reference genes, for the sake of transparency, it should be introduced as part of this study.  
  2. The font size changes abruptly on pages 4 and 8.
  3. Some aspects of the materials and methods should be explained in more detail for the sake of repeatability. The extraction of RNA on line 368 should be fully described rather than referenced.

The use of a 2% agarose gel to inspect quality should be explained in more detail.  RNA gels often contain a denaturing agent such as formaldehyde. Did these gels contain a denaturing agent?

For the cDNA production, lines 372-373, all the reaction components and their concentrations should be listed.

How much cDNA, or at least how much of the cDNA reaction, was used in the qRT-PCR reactions? All the reaction components used for the qRT-PCR reactions should be listed.

  1. I don’t understand the meaning of “Aggregated above two lists…” on lines 441-442 of the conclusions. Do the authors mean they used the two lists? Please clarify.

Author Response

Point 1:  Some of the language in the first paragraph of the introduction should be changed to make it more scientific. The phrase “natural cooking oil” on line 40 is too colloquial. I have no idea what “natural cooking oil” is. Aren’t all cooking oils derived from plant or animal sources, which are “natural”.

The phrase “beneficial for human health” on line 50 is also too colloquial. Lipids are an essential component of the human diet and all may be considered “healthy” since they provide calories. The authors should specifically explain what distinguishes oleic acid (C18:1) from other lipids regarding the human diet?  For example, is it less likely to contribute to heart disease? The unqualified terms “natural” and “healthy” might work for advertisers and marketing companies to sell a product to the general public but they don’t belong in a scientific study without further explanation.  

Response 1: Thanks for your comments and suggestions. The phrases “natural cooking oil” and “beneficial for human health” have been revised on lines 40, 50, and 51 in the revised manuscript.  And four references (No. 11-14) were added to the revised manuscript (on line 51) to support the notion that oleic acid has positive effects on human health.

Point 2: I think the reference to unpublished transcriptome data on line 74 of the introduction belongs somewhere else in the manuscript, perhaps the materials and methods. I also think that, since this data is very important to this study and is the basis of the selection of the reference genes, for the sake of transparency, it should be introduced as part of this study. 

Response 2: We agree with your comments and suggestions. Our transcriptome data have been included as part of this study (on lines 243-246, 281-282, and 450-460), which were deposited at NCBI under the BioProject ID PRJNA703731 and will be released when the article is published.  

Point 3: The font size changes abruptly on pages 4 and 8.

Response 3: Thanks for your comments. We have checked the font size of the full text and changed the font size on pages 2, 4 and 8.

Point 4:     Some aspects of the materials and methods should be explained in more detail for the sake of repeatability. The extraction of RNA on line 368 should be fully described rather than referenced.

The use of a 2% agarose gel to inspect quality should be explained in more detail.  RNA gels often contain a denaturing agent such as formaldehyde. Did these gels contain a denaturing agent?

For the cDNA production, lines 372-373, all the reaction components and their concentrations should be listed.

How much cDNA, or at least how much of the cDNA reaction, was used in the qRT-PCR reactions? All the reaction components used for the qRT-PCR reactions should be listed.

Response 4: Thanks for your comments and suggestions.

The process of RNA extraction has been fully described in the Materials and Methods section (on lines 379-389).

No denaturing agent was added to the 2% agarose gel for inspecting RNA quality.

All the reaction components and their concentration for the cDNA production and the qRT-PCR reactions have been included in the revised manuscript (on lines 394-405).

Point 5: I don’t understand the meaning of “Aggregated above two lists…” on lines 441-442 of the conclusions. Do the authors mean they used the two lists? Please clarify.

Response 5: Thanks for your comments. For clarity, the description was changed to “According to the rank list in the “Tissues of two cultivars” column of Table 5” in the revised manuscript (on lines 469-470).